# Clinical and Biological Characteristics of Four Patients with Aggressive Systemic Mastocytosis Treated with Midostaurin

**DOI:** 10.3390/biomedicines13071655

**Published:** 2025-07-07

**Authors:** Delia Soare, Dan Soare, Camelia Dobrea, Eugen Radu, Horia Bumbea

**Affiliations:** 1Bone Marrow Transplantation Ward, University Emergency Hospital Bucharest, 050098 Bucharest, Romania; delia.soare@drd.umfcd.ro (D.S.); dan.soare@umfcd.ro (D.S.); horia.bumbea@umfcd.ro (H.B.); 2Department of Scientific Research Methodology and Hematology, Carol Davila University of Medicine and Pharmacy, 050098 Bucharest, Romania; 3Center of Hematology and Bone Marrow Transplantation, Fundeni Clinical Institute, 022328 Bucharest, Romania; camelia.dobrea@umfcd.ro; 4Department of Hematology, Fundeni Clinical Institute, Carol Davila University of Medicine and Pharmacy, 022328 Bucharest, Romania; 5Department of Microbiology III, Carol Davila University of Medicine and Pharmacy, 050098 Bucharest, Romania; 6Molecular Pathology Laboratory, University Emergency Hospital Bucharest, 050098 Bucharest, Romania

**Keywords:** aggressive mastocytosis, treatment, midostaurin, C-KIT inhibitor

## Abstract

Systemic mastocytosis (SM) is a rare and heterogeneous disorder characterized by clonal proliferation and accumulation of neoplastic mast cells in one or more organs, most commonly the bone marrow, liver, spleen, and skin. Among its clinical variants, aggressive SM (ASM) presents organ damage and debilitating symptoms due to extensive mast cell infiltration. The management of ASM remains challenging, primarily because treatment must address both symptom control and disease progression. **Background/Objectives**: Recent therapeutic approaches have focused on tyrosine kinase inhibitors (TKIs) that target the oncogenic *KIT* driver mutation, predominantly the D816V mutation, which is implicated in mast cell proliferation. We report a case series of four patients diagnosed with ASM to highlight the real-world experience in the management of ASM. All patients had confirmed *KIT* D816V mutations and presented with signs of advanced organ dysfunction, such as marked hepatosplenomegaly, cytopenia, and significant bone marrow infiltration. First-line therapies, including cytoreductive agents or other TKIs were used. Responses varied in these patients, and ultimately, they were initiated on or transitioned to midostaurin, a multikinase TKI. **Results**: All four patients, after the initiation of midostaurin, presented clinical and biological improvement—at least a clinical improvement response according to the International Working Group-Myeloproliferative Neoplasms Research and Treatment & European Competence Network on Mastocytosis (IWG-MRT-ECNM) criteria. These findings highlight the benefits of KIT inhibition in managing ASM, especially for patients with inadequate responses to traditional therapies. The impact of midostaurin on organ function, mast cell burden, and symptom control emphasizes the importance of the timely integration of TKIs into therapeutic protocols. However, optimal treatment duration, long-term safety, and the development of acquired resistance remain critical questions that warrant further studies. Larger prospective trials are needed to better delineate the prognostic factors associated with sustained response, refine patient selection, and explore combination strategies that may enhance therapeutic efficacy. **Conclusions**: The patients presented in this case series benefited from midostaurin therapy, showing either a clinical improvement or partial response according to the IWG-MRT-ECNM criteria. Our case series illustrates that KIT inhibitors can offer meaningful clinical benefit in ASM, reinforcing their position as an emerging cornerstone option in ASM management.

## 1. Introduction

Systemic mastocytosis (SM) is a rare hematologic disorder characterized by abnormal mast cell accumulation in multiple organs, including bone marrow, liver, spleen, and skin [1]. Typically, secondary to mutations in the *KIT* gene, SM can manifest with a broad range of clinical pictures, from indolent to aggressive subtypes. Symptoms may include allergic reactions, cutaneous lesions, gastrointestinal disturbances, and musculoskeletal pain. Persistent mast cell activation can cause organ dysfunction, posing a significant morbidity and mortality risk. Diagnostic approaches involve histopathological and molecular assays, while management includes symptom control, targeted therapies, and supportive interventions. Emerging treatments aim to improve long-term outcomes for affected individuals [2].

Although systemic mastocytosis is classified as a rare disease, its impact at the population level is not negligible. Recent nationwide data from Sweden report an annual incidence of approximately 1.6 cases per 100,000 adults and a prevalence approaching 24 per 100,000 inhabitants [3]. Similar figures have been observed in regional studies from Italy, where the incidence is estimated at 1.1 and the prevalence ranges between 10 and 17 per 100,000 [4]. Epidemiological patterns and clinical phenotypes appear to vary by geographic region; however, Central and Eastern Europe remain insufficiently represented in large-scale datasets [3,5,6]. Although considered a rare disease, its population burden is non-trivial: recent nationwide data from Sweden place the annual adult incidence at ≈1.6 per 100,000 and the prevalence near 24 per 100,000 [3], while regional Italian estimates are comparable (incidence 1.1 and prevalence 10–17 per 100,000) [4]. Geography appears to influence both epidemiology and phenotype, yet Central/Eastern Europe remains under-represented in large datasets [3,5,6]. Consequently, precise epidemiological data in individual countries are limited; for example, the true prevalence in Romania is not well documented, and only a very small number of patients have been recorded to date [7]. Midostaurin (PKC412; N-benzoyl staurosporine) is a semi-synthetic indolocarbazole originally isolated from *Streptomyces staurosporeus* and first described as a potent protein-kinase-C antagonist [8]. Subsequent kinase-profiling revealed a far broader spectrum encompassing class III receptor tyrosine kinases, notably FLT3 and KIT, including the imatinib-refractory *KIT* D816V mutation that drives SM [9]. In neoplastic mast cells, midostaurin binds the ATP-binding pocket of KIT, FLT3, and allied kinases, thereby extinguishing proliferative and pro-survival signaling cascades and simultaneously curtailing FcεRI-dependent degranulation [10]. This dual cytostatic and anti-secretory action translates into tangible clinical benefit: Phase II trials in advanced SM have reported overall response rates of 60–75%, durable reductions in serum tryptase and splenomegaly, and median survivals extending beyond two years even in aggressive subtypes [11]. Notably, oral administration (100 mg bid) is generally well tolerated, and midostaurin has become the first disease-modifying therapy licensed for aggressive or refractory mast cell disease, shifting the management paradigm from purely symptomatic control toward targeted, survival-modifying intervention [12].

This case series aims to present the experience of the largest known cohort of patients with aggressive systemic mastocytosis (ASM) treated with midostaurin in Romania. Given the rarity of the disease and the limited access to targeted therapies in our country, our objective was to contribute real-world data from a national reference center with expertise in mastocytosis. Furthermore, this study addresses a gap in the literature by reporting outcomes from a region where data on ASM management are scarce, with the ultimate goal of supporting future therapeutic decision-making and encouraging broader access to advanced treatments for rare hematologic malignancies.

## 2. Materials and Methods

The diagnosis of SM was made according to the World Health Organization (WHO) guidelines used at the time of the diagnosis and confirmed according to the current WHO guidelines and International Consensus Classification (ICC) recommendations. And the response criteria were assessed using the International Working Group-Myeloproliferative Neoplasms Research and Treatment & European Competence Network on Mastocytosis (IWG-MRT-ECNM) Response Criteria for Advanced Mastocytosis [13].

All patients were referred to our multidisciplinary outpatient clinic for mastocytosis evaluation, and suspected SM underwent comprehensive diagnostic evaluation, including bone marrow (BM) smear and biopsy and detection of *KIT* D816V mutation. Informed consent was obtained according to the guidelines of the local ethics committee.

Information about the clinical symptoms was obtained from the medical electronic charts as well as from medical interviews of the participants. The prognostic score was assessed using the International Prognostic Scoring System for Advanced Systemic Mastocytosis, based on data available at the time of diagnosis in the electronic medical chart [13].

## 3. Results

Below, we present the four clinical cases. The baseline characteristics of all four patients are summarized in Table 1.

Case 1

A 57-year-old female patient diagnosed with ASM, fulfilling one major and three minor diagnostic criteria, with 55% bone marrow infiltration. She underwent 2 cycles of cladribine, which yielded a clinical response, followed by rapid disease progression after discontinuation of cladribine. Since midostaurin or a clinical trial was not available in Romania at that time, she received imatinib for 6 months. Also, the patient required repeated paracentesis secondary to portal hypertension with ascites secondary to hepatomegaly. Her complete blood count (CBC) showed marked leukocytosis (maximum value 25,600/μL), which did not improve under imatinib.

Conversely, targeted therapy with midostaurin was started, and she became the first patient in Romania to receive this agent. Under midostaurin, she achieved a rapid clinical and biological response, documented by the resolution of ascites, splenic size reduction, decreased leukocyte count, improvement in quality of life, and over 50% decreases in both serum tryptase levels and bone marrow mast cell infiltration. The patient experienced dyspeptic syndrome with severe nausea as a side effect of midostaurin, which was managed with 5-HT_3_ receptor antagonists. According to the IWG-MRT-ECNM Response Criteria [13], the patient achieved partial remission.

Case 2

A 44-year-old female was diagnosed with ASM following a biopsy of a vertebral osteolytic lesion, similar to previously reported cases [13,14,15,16]. She had been experiencing severe bone pain for approximately two years prior to diagnosis. She underwent several MRIs and PET-CT scans, documented multiple osteolytic non-FDG-avid lesions with a maximum size of 18 mm on T8 vertebra, and other lesions localized on the T10 vertebra (15 mm in diameter), and on the L3 vertebra 14 mm in diameter). At first, she underwent investigations for multiple myeloma or other neoplastic diseases with secondary bone lesions without any certain diagnostic criteria. After 2 years of investigations, she was referred to our center, and a biopsy harvested from an osteolytic lesion proved an infiltration with atypical mastocytes positive for CD25 and CD138. The biopsy also documented bone marrow infiltration of over 50% atypical mastocytes with marked osteosclerosis. Midostaurin therapy was initiated alongside supportive treatment with bisphosphonates and later anti-RANKL therapy. The outcome of the treatment was marked by clinical improvement characterized by decreased tryptase levels, but not over 50%, consolidation of osteolytic lesions, and resolution of bone pain. The patient experienced dyspeptic syndrome with severe nausea as a side effect of midostaurin, which was managed with 5-HT_3_ receptor antagonists. The patient followed successful monotherapy with midostaurin for over 3 years, and her bone marrow infiltration is now 15% with atypical clonal mastocytes. According to the IWG-MRT-ECNM Response Criteria [13], the patient achieved partial remission.

Case 3

In the specialized literature, it is reported that peripheral adenopathy occurs in 26% of cases, whereas central adenopathy is observed in 19% [17]. The retroperitoneal, mesenteric, periportal, and axillary regions represent the most frequently affected lymph node sites [17,18,19,20]. A 75-year-old male patient presented with multiple lymphadenopathies with maximum axial dimension 23/30 mm, hepatomegaly, splenomegaly, ascites, and normochromic normocytic mild anemia (10 g/dL hemoglobin). The patient was investigated in another center where a lymph node biopsy confirmed mast cell infiltration, and a bone marrow biopsy showed approximately 20% mast cell infiltration. Specific treatment with midostaurin was initiated, leading to the resolution of lymphadenopathies and reduction of splenomegaly. However, due to cardiac comorbidities and the associated dyspeptic syndrome, he was unable to tolerate the full dose; hence, he received 50 mg every 12 h and still achieved a clinical improvement response according to the IWG-MRT-ECNM Response Criteria [8]. Consequently, the recommended management was to continue low-dose therapy until loss of clinical response. The patient maintains the same clinical response, as confirmed at the most recent follow-up conducted in May 2025.

Case 4

Smoldering systemic mastocytosis (SSM) may carry a higher risk of disease progression and leukemic transformation, estimated at around 18%, compared to other subtypes of indolent systemic mastocytosis (ISM) [15]. This elevated risk underscores the importance of close clinical monitoring as SSM patients can experience more significant organ involvement and a greater mast cell burden than those with other ISM variants. Consequently, timely recognition of disease evolution is critical for optimizing patient management and improving long-term outcomes [21,22]. A 53-year-old female patient had been under medical observation for approximately six years, initially presenting to the Allergy Department. Working as a florist, she developed allergic reactions to pollen, prompting measurement of her serum tryptase, which was found to be elevated at 168 µg/L. Further investigations revealed 45% bone marrow infiltration with atypical mastocytes and associated hepatosplenomegaly. A diagnosis of SSM was established, and cladribine therapy was initiated. She underwent one cycle of cladribine, leading to clinical improvement.

Five years later, her disease progressed; she presented with a deteriorated general condition, severe bone pain, and weight loss. After upper gastrointestinal endoscopy, the patient was diagnosed with gastrointestinal mast cell infiltration. The disease advanced to ASM, and midostaurin treatment was initiated. Six months after starting therapy, a clinical improvement response was observed according to the IWG-MRT-ECNM Response Criteria for Advanced Systemic Mastocytosis [13]. The patient presented with dyspeptic syndrome and diarrhea following administration of TKI therapy, which were managed with the antiemetic granisetron and the antidiarrheal loperamide.

## 4. Discussion

Midostaurin, a multikinase inhibitor with potent activity against *KIT* D816V, has emerged as a key therapeutic option in advanced systemic mastocytosis, including ASM, SM with an associated hematologic neoplasm, and mast cell leukemia. Traditionally, treatments for ASM were limited and primarily aimed at symptom control with agents such as H1/H2 antihistamines, corticosteroids, and cytoreductive therapies (e.g., interferon-alpha, cladribine) [23]. However, the approval of midostaurin represented a major breakthrough in the management of ASM, backed by robust clinical evidence [12].

A pivotal, multicenter Phase II trial [6] evaluating midostaurin in ASM demonstrated significant clinical benefits, including reductions in mast cell burden, improvements in organ function, and alleviation of symptoms. Overall response rates exceeded 50%, with a subset of patients achieving major responses evidenced by decreased splenomegaly and improved peripheral blood counts [12]. Additional analyses demonstrated that midostaurin was associated with improved overall survival compared to historical controls, highlighting its potential to alter the disease course.

The cases described above provide real-world insights into midostaurin’s efficacy and tolerability. In Case 1, the patient with ASM experienced rapid clinical and biological improvement after other options (cladribine, imatinib) failed to control her disease. Notably, ascites was resolved, splenic size decreased, and leukocytosis was stabilized, signifying both hematologic and organ-level responses. These results echoed published findings that midostaurin can induce rapid control of mast cell proliferation and mitigate complications such as massive splenomegaly [11,12,14].

In Case 2, the patient with vertebral osteolytic lesions and intractable bone pain was characterized by relief of symptoms following a combination therapy with midostaurin and bisphosphonates, underscoring the agent’s ability to reduce mast cell-mediated bone involvement. Similarly, Case 3 demonstrated the capacity of midostaurin to reduce nodal disease and splenomegaly in a patient with multiple comorbidities. Although dose adjustments were necessary in Case 3 due to cardiac and gastrointestinal side effects, the patient still attained a positive clinical response, underlining midostaurin treatment feasibility even in older or less fit patients, when carefully monitored and with modified doses.

Finally, Case 4 illustrated disease progression from SSM to an aggressive form, with midostaurin providing a partial response within six months of treatment. Despite side effects such as dyspepsia and diarrhea—common toxicities documented in clinical trials—symptomatic relief was achieved through supportive medications (antiemetics and antidiarrheals). This real-world scenario aligns with data indicating that gastrointestinal side effects are among the most frequently encountered adverse events but can be managed successfully to maintain dose intensity [23].

At the molecular level, midostaurin acts as a type-I kinase inhibitor, stabilizing KIT and FLT3 in an inactive conformation [24]. In *KIT*-D816V-positive Ba/F3 and HMC-1.2 cells, nanomolar midostaurin exposure abrogates autophosphorylation of the receptor and suppresses downstream STAT5, PI3K/AKT, MAPK, and FES activity [25]. The resultant signaling collapse triggers G_1_ cell-cycle arrest through downregulation of cyclin-D2 and upregulation of the CDK inhibitor p27^Kip1^, followed by mitochondrial apoptosis characterized by BAX translocation, cytochrome-c release, and caspase-9/3 activation [26,27]. Notably, the drug also retains equivalent potency against wild-type KIT and the *KIT* V560G juxtamembrane domain mutant, underscoring its versatility across mast cell neoplasms [12]. Synergistic lethality has been documented with dasatinib [28], cladribine, and BH3-mimetics [29,30,31], reflecting complementary inhibition of BTK, LYN, or BCL-2 family proteins and defining rational combination strategies for refractory disease [8].

Pharmacokinetic studies identify two active midostaurin metabolites—CGP 52421 and CGP 62221—with half-lives exceeding that of the parent compound. CGP 62221 retains anti-proliferative activity against KIT-dependent mast cells, whereas both metabolites inhibit IgE-triggered degranulation, suggesting sustained anti-secretory coverage during chronic dosing. Importantly, midostaurin does not cause global immunosuppression, and lymphopenia and immunoglobulin levels remain stable, mitigating infection risk in a population often rendered cytopenic by marrow infiltration [32]. Early evidence further indicates modulation of the cytokine milieu: treated patients exhibit declining serum IL-6 and CCL2 [33], mediators implicated in osteolysis and constitutional symptoms [34]. Whether such microenvironmental reprogramming potentiates the drug’s direct tumoricidal effect or leads to resistance through adaptive stromal signaling remains under active investigation [35].

Although ~90–95% of adult SM patients carry a *KIT* mutation at codon 816 (most often D816V), a minority harbor non-D816V *KIT* variants [34]. Examples include other exon 17 activation loop mutations (e.g., D816Y, D816H, D815K, D820G) as well as alterations in the juxtamembrane domain (exons 8–11, such as V560G or internal tandem duplications) [36]. The presence of these atypical mutations can significantly influence therapeutic choices and response. Midostaurin is a type I kinase inhibitor that can bind the active conformation of KIT, and it retains activity against many activation-loop mutants. In vitro, midostaurin effectively inhibits *KIT* D816V as well as other codon 816 substitutions like D816Y [36]. Clinically, midostaurin has shown efficacy in advanced SM patients regardless of *KIT* mutation status—the pivotal trial included a small subset of D816V-negative cases and still achieved ~60% response rates overall [9,37].

That said, the type of *KIT* mutation can guide the choice of kinase inhibitor. Mutations in exons 8–11 (outside the *KIT* catalytic domain) typically confer sensitivity to imatinib, a different TKI. Unlike D816V, these juxtamembrane domain mutations do not lock KIT in the active conformation and, therefore, do not interfere with imatinib [38,39,40,41,42,43]. In such cases (often seen in well-differentiated SM variants), imatinib has achieved complete and durable remissions [38,39,40,41,42,43]. Consensus recommendation: in a confirmed *KIT* D816V-negative patient, the entire *KIT* gene should be sequenced to identify the mutation—if a lesion in exon 8–11 is found, imatinib can be chosen, whereas a mutation in exon 17 (even if not D816V) predicts imatinib failure and thus supports using midostaurin or another D816V-active inhibitor [9].

Co-mutations in *SRSF2*, *ASXL1*, and *RUNX1* profoundly influence the course of ASM under midostaurin therapy, being correlated with lower response rates, shorter progression-free intervals, and decreased overall survival on midostaurin [44,45,46]. Mechanistically, they reflect a more complex, KIT-independent oncogenic network, explaining relative drug resistance [44,45]. These insights have been translated into risk stratification (e.g., the S/A/R panel defines high-risk disease) and inform management by flagging patients who may need earlier aggressive therapy or transplant despite midostaurin [47]. Importantly, even S/A/R-mutated patients can benefit from midostaurin (as evidenced by extended survival compared to historical outcomes) [44,47], but expectations must be tempered and therapy optimized. Ongoing trials and newer KIT inhibitors are being explored to overcome this high-risk biology. In practice, an advanced SM patient with *SRSF2*, *ASXL1*, or *RUNX1* mutations is treated as a high-risk case: midostaurin is initiated, but the care team remains vigilant for inadequate response, considers combination therapies, and often plans a definitive curative approach (transplant) if feasible. This personalized strategy, guided by co-mutation status, aims to improve outcomes in the era of targeted therapy for mastocytosis.

Midostaurin is generally well tolerated; however, gastrointestinal (GI) adverse events are among the most frequently reported toxicities. In pivotal trials, nausea (up to 82%), vomiting (up to 60%), and diarrhea (up to 43%) were the most common side effects observed, particularly during the initial weeks of treatment [48]. In our case series, the majority of patients (3 out of 4) developed gastrointestinal symptoms consistent with dyspeptic syndrome, including nausea and vomiting. These adverse effects were effectively managed with supportive antiemetic therapy, including granisetron and ondansetron, allowing patients to continue treatment without dose interruptions. Regarding hematologic toxicity, grade 3–4 cytopenia, such as neutropenia and thrombocytopenia, have been reported in up to 20–30% of patients treated with midostaurin [48,49]. In our cohort, no severe hematologic toxicity requiring dose modification was observed. Hepatotoxicity is less common, with elevated transaminases occurring in <10% of cases and usually being transient and manageable [13,30]. None of the patients in our series developed clinically significant liver function abnormalities during treatment.

Beyond direct cytotoxicity, midostaurin also exerts a profound influence on mast cell functions. Micromolar concentrations almost completely abolish FcεRI-mediated histamine, leukotriene, and cytokine release [10] from primary human mast cells and basophils, an effect attributed to simultaneous inhibition of PKC, SYK, and LYN—critical early steps in the IgE-receptor activation cascade [8,24,35,37,41,50]. In ex-vivo setting, basophils from SM patients exposed to midostaurin, histamine release was attenuated, correlating with rapid amelioration of pruritus, flushing, and anaphylactoid episodes [12]. By curbing the secretion of vasoactive amines, proteases, and pro-angiogenic factors, the drug may influence the re-shaping of the perivascular microenvironment, dampening endothelial activation and fibroblast proliferation that otherwise sustain fibrosis and organomegaly in advanced SM [8,50,51].

Overall, midostaurin has become a mainstay therapy in ASM due to its targeted mechanism against various mutant KIT receptors [36]. Clinical observations and accumulated evidence indicate that midostaurin not only reduces mast cell burden but also significantly improves quality of life by controlling disease-related symptoms, organomegaly, and cytopenias [12]. Close monitoring is essential, particularly for patients prone to cardiac or gastrointestinal complications, emphasizing the importance of multidisciplinary care.

Despite demonstrating clinical efficacy in patients with ASM, midostaurin has several well-recognized limitations. While it can induce partial or even complete responses in a subset of patients, the majority experience only temporary disease control, with eventual progression observed over time [44]. Complete remissions are rare, and even when achieved, they are not sustained in most cases [44]. Furthermore, midostaurin does not eradicate the underlying clonal hematopoietic stem cell population harboring the *KIT* D816V mutation and, therefore, cannot be considered curative [11,12,44]. Its efficacy is further reduced in patients with co-occurring high-risk mutations (e.g., *SRSF2*, *ASXL1*, *RUNX1*), which are associated with lower response rates and inferior survival outcomes, highlighting the heterogeneity and molecular complexity of the disease [11,44,49].

Another key consideration is the timing of treatment initiation. Due to the rarity of the disease and its frequently indolent early course, systemic mastocytosis is often diagnosed at a late stage, when symptoms have progressed and organ damage has already occurred. This diagnostic delay, compounded by the lack of awareness and access to molecular testing, often precludes the initiation of targeted therapy at the actual onset of disease. Consequently, while earlier intervention with KIT inhibitors such as midostaurin could hypothetically offer greater benefit by preventing disease progression, in real-world clinical practice, such early initiation is rarely feasible. This reflects a broader challenge in the management of rare hematologic malignancies, where optimal treatment timing may be constrained by diagnostic and systemic factors [47,52].

## Figures and Tables

**Table 1 biomedicines-13-01655-t001:** Baseline characteristics at diagnosis and treatment response criteria.

	Case 1	Case 2	Case 3	Case 4
Age at diagnosis	57	44	75	53
Sex	Female	Female	Male	Female
Serum tryptase level at diagnosis ng/mL	387	319	64.6	208
*KIT* D816V mutation	Positive	Positive	Positive (VAF 7.89%)	Positive
Bone marrow mast cell burden	40%	55%	20%	55%
Bone marrow fibrosis	MF-3	MF-2	Not evaluated	MF-3
ALK at diagnosis	256	110	114	79
ALK at midostaurin initiation	386	192	149	108
Hepatomegaly	Yes	No	Yes	Yes
Splenomegaly	Yes	No	Yes	Yes
B findings (mast cell burden/organ infiltration)	Hepatomegaly, splenomegaly, signs of dysplasia or myeloproliferation, 40% mast cells in bone marrow, and serum total tryptase 387 ng/mL	50% mast cells on bone marrow and serum tryptase of 319 ng/mL	Lymphadenopathy, hepatomegaly, splenomegaly 45% mast cells on bone marrow biopsy	Hepatomegaly, splenomegaly, signs of dysplasia or myeloproliferation, 55% mast cells on bone marrow biopsy, and serum tryptase of 208 ng/mL
C findings (organ dysfunction)	Hepatomegaly with ascites, portal hypertension, malabsorption with weight loss	Skeletal involvement	Bone marrow dysfunction caused by neoplastic mast cell infiltration, and malabsorption with weight loss.	Hepatomegaly with impairment of liver function, ascites, malabsorption with weight loss
Treatment period (follow-up)	57 months	42 months	22 months	9 months
Previous treatments	Cladribine, imatinib	None	None	Cladribine
Prognostic score at diagnosis (IPSM)	1	1	3	1
Midostaurin Treatment Response IWG-MRT-ECNM response criteria	Partial remission	Clinical improvement	Clinical improvement	Clinical improvement
Organ function improvement and symptom relief after midostaurin	Resolution of ascites, splenic size reduction over 50%, decrease in both serum tryptase and bone marrow mast cell infiltration, resolution of malabsorption	Decreased tryptase levels, consolidation of osteolytic lesions, and resolution of bone pain	Resolution of lymphadenopathies, splenic size reduction	Resolution of bone pain, resolution of malabsorption
Midostaurin-related side effects	Nausea	Nausea	Nausea	Nausea and diarrhea

Abbreviations: ALK—alkaline phosphatase, IPSM—international prognostic scoring system for mastocytosis, IWG-MRT-ECNM—International Working Group on Myeloproliferative Neoplasms Research and Treatment & European Competence Network on Mastocytosis, MF—myelofibrosis grade, VAF—variant allele frequency.

## Data Availability

The original contributions presented in this study are included in the article. Further inquiries can be directed to the corresponding author.

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
