# Peer review of "Clinical and Biological Characteristics of Four Patients with Aggressive Systemic Mastocytosis Treated with Midostaurin"

_biomedicines, 2025, doi:10.3390/biomedicines13071655_

Round 1

Reviewer 1 Report

Comments and Suggestions for Authors

Soare D et al show their experience in the management of systemic mastocytosis in 4 patients affected by this disease. Overall, the article shows data that, together with the other existing data, may contribute to an in-depth understanding of the impact of Midostaurin as an optional treatment for systemic mastocytosis, especially when the KIT D816V mutation is present.

However, some minor concerns arose for me after reading the article:

(a) Throughout the introduction, the authors only mention that DM is a rare disease; it would be interesting if they included some statistical data on its prevalence in Europe and in the study country, comparatively, as well as the risk factors associated with this disease.

b) Their data suggest that the use of Midostaurin has a beneficial effect on the quality of life of patients receiving these drugs; however, in none of the cases studied, I am aware that there were only 4, was a clearance of the disease found, however the authors only limit their discussion to the partial improvement in the state of the disease, but do not discuss the possible scenario in which from the onset of the disease patients receive midostaurin, or what are the limitations of therapy with these drugs.

c)  Could the patients' table be improved?

d) AKT= anaplastic lymphoma kinase ??? Please include it in the list of abbreviations.

e) Are you sure that the dyspeptic syndrome is a consequence of the treatment and not the disease? How do you know, if a healthy control was not included?

f) What is the impact of the other mutations that have been described in MS on the effects of midostaurin?

Author Response

Regarding the reviewer's comments and sugestions.

Comment (a) Throughout the introduction, the authors only mention that DM is a rare disease; it would be interesting if they included some statistical data on its prevalence in Europe and in the study country, comparatively, as well as the risk factors associated with this disease.

Response comment (a): We thank the reviewer for this insightful comment. We added a new paragrap in the introduction section (page 2, line 63), in which we present the general epidemiology for SM. Unfortunantely, the prevalence and incidence in Romania for SM is not well documented.

Comment (b) Their data suggest that the use of Midostaurin has a beneficial effect on the quality of life of patients receiving these drugs; however, in none of the cases studied, I am aware that there were only 4, was a clearance of the disease found, however the authors only limit their discussion to the partial improvement in the state of the disease, but do not discuss the possible scenario in which from the onset of the disease patients receive midostaurin, or what are the limitations of therapy with these drugs.

Response (b): We fully agree that the potential timing of midostaurin initiation and the limitations of this therapeutic approach are important aspects to consider. As systemic mastocytosis is a rare disease, it is often associated with delayed diagnosis, which frequently precludes the possibility of initiating targeted therapy at disease onset. Furthermore, while midostaurin has demonstrated the capacity to induce partial or even complete remission in some cases, it does not offer a definitive cure for this chronic and incurable condition. We have now addressed these considerations in the revised Discussion section, acknowledging both the therapeutic potential and the current limitations of midostaurin in ASM management. These sections are present on page 8, starting with the paragraph starting from line 347 and also the next paragraph which starts at line 358.

Comment (c) Could the patients' table be improved?

Response - the patient table was completely modified, we also added new sections in which we presented specifficaly what biological and clinical parameters were improved after midostaurin initiation, and also added a section with side effects observed after midostaurin intiation. The table is now found at page 6.

Comment (d) AKT= anaplastic lymphoma kinase ??? Please include it in the list of abbreviations.

Response (d) We thank the reviewer for their observation. We would like to clarify that AKT is already included in the list of abbreviations; however, we acknowledge that the current expanded form may have caused confusion. Indeed, in the context of the PI3K/AKT signaling pathway, AKT refers to "protein kinase B" and not to "anaplastic lymphoma kinase." We will revise the abbreviation explanation accordingly in the manuscript to ensure greater clarity and consistency with standard usage in the field. The correct abbreviation is found at page 10, in the abbreviation table.

Comment (e)  Are you sure that the dyspeptic syndrome is a consequence of the treatment and not the disease? How do you know, if a healthy control was not included?

Response (e) 

We thank the reviewer for raising this important point. In the absence of a control group, we acknowledge the limitations in establishing a definitive causal relationship. However, we would like to clarify that none of the patients included in our cohort reported symptoms consistent with dyspeptic syndrome prior to the initiation of midostaurin therapy. Given the temporal association and the onset of symptoms following treatment initiation, these gastrointestinal manifestations were considered likely adverse reactions to the drug.

This interpretation is also supported by data from the literature, which consistently report nausea and vomiting as among the most common adverse effects of midostaurin. According to the summary of product characteristics and clinical trial data, gastrointestinal symptoms—including nausea (79%), vomiting (66%), and diarrhea (54%) are frequently observed in patients receiving Midostaurin, particularly during the first weeks of treatment. We addressed this by adding a paragraph in the dicussions section in which we describe the side effects of midostaurin administration - page 8, line 316.

Comment (f) What is the impact of the other mutations that have been described in MS on the effects of midostaurin?

Response (f): We addressed this aspect by adding two paragraphs in the discussions section in which we described the effect of midostaurin on non-D816V mutations (page 8, line 277 and line 288). Also we added a paragraph in which we describe the effect of the presence of co-mutations in SRSF2, ASXL1, and RUNX1 and the impact of midostaurin tratement (page 8, line 298)  

Reviewer 2 Report

Comments and Suggestions for Authors

This case series study on advanced systemic mastocytosis (ASM) explores the potential clinical value of KIT inhibitors (midostaurin). However, the lack of key data undermines the credibility of its conclusions:

Missing clinical data:

Although four ASM patients were mentioned, the study fails to provide baseline characteristics , laboratory parameters (e.g., serum tryptase levels, bone marrow mast cell burden percentage), or imaging findings (e.g., liver/spleen size measurements).

The term "limited or transient response" is not quantified—was it partial response (PR) or stable disease (SD)? What criteria were used (e.g., IWG-MRT criteria)?

Insufficient treatment response data:

The efficacy of midostaurin is vaguely described (e.g., "encouraging effects") without objective measures:

Organ function improvement (e.g., changes in ALT/AST, reduction in spleen volume);

Symptom relief (e.g., frequency of mast cell activation episodes, changes in VAS pain scores);

Survival outcomes (e.g., progression-free survival [PFS], overall survival [OS]).

Lack of safety data:

No adverse events (e.g., hematologic toxicity, hepatotoxicity incidence) were reported, which is critical for assessing the risk-benefit ratio.

Additional Revision Requests

Abstract:

The authors claim that "these findings highlight the importance of KIT inhibition in ASM management," but no midostaurin treatment outcomes are provided in the abstract. Please include supporting data.

Introduction:

The first-line treatments for ASM should be briefly introduced before stating that "some patients show only limited or transient responses."

The last paragraph of the Introduction should clearly state the research gap this study aims to address.

Formatting Issues:

Lines 77 & 86: References should be placed before the period (e.g., "…as previously described[1]." → "…as previously described [1].").

Line 81: A sentence ends without punctuation—please correct.

Author Response

Comment 1: Although four ASM patients were mentioned, the study fails to provide baseline characteristics , laboratory parameters (e.g., serum tryptase levels, bone marrow mast cell burden percentage), or imaging findings (e.g., liver/spleen size measurements).

Response: 

We would like to respectfully clarify that the clinical and biological baseline characteristics, including laboratory parameters (such as serum tryptase levels and bone marrow mast cell burden) and relevant imaging findings (presence or absence of hepato-/splenomegaly, lymphadenopathies, bone lesions), are indeed presented in the manuscript. These data are provided both within the individual case descriptions in the main text and summarized collectively in Table 1 (page 6), which was designed to present the baseline clinical features of all four ASM patients in a concise format.

To improve clarity, we will revise the table title and corresponding text to more explicitly reflect the inclusion of these baseline characteristics (page 4, line 128). The tabel title was changed to: "Table 1 Baseline characteristics at diagnosis and treatment-response criteria" (page 6)

Comment 2: The term "limited or transient response" is not quantified—was it partial response (PR) or stable disease (SD)? What criteria were used (e.g., IWG-MRT criteria)?

Response 2: Following your observation, we have removed the vague expression "limited or transient response" from the abstract, as it lacked precision and could be misleading. The clinical course and treatment response for each individual patient are now described more clearly within the main text of the manuscript.

Due to heterogeneity in treatment approaches, including suboptimal therapy in some patients and monotherapy with midostaurin, it is challenging to uniformly classify treatment response across all cases at the initiation of midostaurin. Moreover given condensed format of the abstract, we felt it was more appropriate to eliminate the non-specific terminology rather than attempt to condense a nuanced clinical discussion.

Comment 3: Insufficient treatment response data:

The efficacy of midostaurin is vaguely described (e.g., "encouraging effects") without objective measures:

Organ function improvement (e.g., changes in ALT/AST, reduction in spleen volume);

Symptom relief (e.g., frequency of mast cell activation episodes, changes in VAS pain scores);

Survival outcomes (e.g., progression-free survival [PFS], overall survival [OS]).

Response 3: We would like to clarify that the treatment response to midostaurin was described for each patient individually in the main text of the manuscript, including the evolution of organ involvement and symptomatology where relevant. However, in order to improve the clarity and readability of the article, and also considering a related suggestion from another reviewer regarding the modification of Tabel 1, we have revised the table to include two additional rows:
(1) one summarizing the type of treatment response observed following midostaurin therapy, and
(2) another indicating which of the B and/or C findings showed improvement during follow-up.

Regarding symptom relief, none of the patients experienced mast cell activation or degranulation episodes at any point, and therefore this aspect was not discussed. The only patients who reported bone pain experienced improvement under treatment, and this was explicitly mentioned in the Table.

As also noted in the manuscript, the response to midostaurin has been maintained in all four cases so far. However, given the very small number of patients included in this case series, standard outcome measures such as overall survival (OS) or progression-free survival (PFS) are not appropriate or statistically meaningful and were therefore not reported.

We appreciate the reviewer’s suggestion, which helped us improve the structure and clarity of the manuscript.

Comment 4: Lack of safety data:

No adverse events (e.g., hematologic toxicity, hepatotoxicity incidence) were reported, which is critical for assessing the risk-benefit ratio.

Response 4: Indeed, while adverse events were briefly mentioned at the end of each individual case description in the main text, we acknowledge that hematologic toxicity was not explicitly addressed. In response to your valuable suggestion and recognizing the importance of this aspect in assessing the risk-benefit profile of midostaurin, we have revised the safety paragraph in the Discussion section (page 8, starting at line 314). Specifically, beginning with line 317, we have now included a dedicated comment on the adverse event profile observed in our case series. We clearly state that none of the four patients developed cytopenias during treatment (page 9, line 324), and we have expanded on the tolerability of the regimen based on our observations.

Comment 5: Abstract: The authors claim that "these findings highlight the importance of KIT inhibition in ASM management," but no midostaurin treatment outcomes are provided in the abstract. Please include supporting data.

To provide the important clinical context to support our claim, at the begining of the results section of the from the abstract we added the following phrase (line 29): All four patients, after the initiation of midostaurin, presented clinical and biological improvement – at least a clinical improvement response according to the International Working Group-Myeloproliferative Neoplasms Research and Treatment & European Competence Network on Mastocytosis (IWG-MRT-ECNM) criteria.

Comment 6:  The first-line treatments for ASM should be briefly introduced before stating that "some patients show only limited or transient responses."

Response 6: Thank you for this valuable comment. We note that another reviewer also found the wording “some patients show only limited or transient responses” too vague. To address this concern, we have removed that phrase and revised the Abstract as follows (line 26): “First-line therapies, including cytoreductive agents or other TKIs, were used. Responses varied in these patients, and ultimately, they were initiated or transitioned to midostaurin, a multikinase TKI.” Additionally, in the section where each case is described in detail, the exact cytoreductive treatment and the specific TKI administered to each patient is mentioned. We believe these clarifications provide the necessary context for understanding why certain patients exhibited limited or transient responses.

Comment 7: The last paragraph of the Introduction should clearly state the research gap this study aims to address.

Response 7: We thank the reviewer for this helpful observation. In response, we have added a concluding paragraph to the Introduction (page 2, line 100) section that explicitly outlines the research gap this case report aims to address. We hope this addition provides the necessary clarification regarding the study’s purpose and its contribution to the current literature.

Other comments: Formatting Issues:

Lines 77 & 86: References should be placed before the period (e.g., "…as previously described[1]." → "…as previously described [1].").

Line 81: A sentence ends without punctuation—please correct.

Response: Errors were corrected

Round 2

Reviewer 2 Report

Comments and Suggestions for Authors

Null